# Antibody Response against SARS-CoV-2 after mRNA Vaccine in a Cohort of Hospital Healthy Workers Followed for 17 Months

**DOI:** 10.3390/vaccines12050506

**Published:** 2024-05-07

**Authors:** Domenico Tripodi, Roberto Dominici, Davide Sacco, Claudia Pozzobon, Simona Spiti, Rosanna Falbo, Paolo Brambilla, Paolo Mascagni, Valerio Leoni

**Affiliations:** 1Laboratory of Clinical Pathology and Toxicology, Hospital Pio XI of Desio, ASST-Brianza, 20832 Desio, Italy; domenicotripodi96@gmail.com (D.T.); roberto.dominici@asst-brianza.it (R.D.);; 2Department of Medicine and Surgery, University of Milano-Bicocca, 20900 Monza, Italy; 3Department of Brain and Behavioural Sciences, Università degli Studi di Pavia, 27100 Pavia, Italy; davide.sacco02@universitadipavia.it; 4Laboratory of Medical Genetics, Centro Diagnostico Italiano, 20100 Milan, Italy; 5Clinical Unit of Occupational Health, Desio Hospital, ASST Brianza, 20832 Desio, Italy

**Keywords:** COVID-19, antibody response, mRNA vaccine, omicron variant infection, healthcare workers (HCWs)

## Abstract

The assessment of antibody response to severe acute respiratory syndrome coronavirus 2 (SARS-CoV-2) is of critical importance to verify the protective efficacy of available vaccines. Hospital healthcare workers play an essential role in the care and treatment of patients and were particularly at risk of contracting the SARS-CoV-2 infection during the pandemic. The vaccination protocol introduced in our hospital protected the workers and contributed to the containment of the infection’ s spread and transmission, although a reduction in vaccine efficacy against symptomatic and breakthrough infections in vaccinated individuals was observed over time. Here, we present the results of a longitudinal and prospective analysis of the anti-SARS-CoV-2 antibodies at multiple time points over a 17-month period to determine how circulating antibody levels change over time following natural infection and vaccination for SARS-CoV-2 before (T0–T4) and after the spread of the omicron variant (T5–T6), analyzing the antibody response of 232 healthy workers at the Pio XI hospital in Desio. A General Estimating Equation model indicated a significant association of the antibody response with time intervals and hospital area, independent of age and sex. Specifically, a similar pattern of antibody response was observed between the surgery and administrative departments, and a different pattern with higher peaks of average antibody response was observed in the emergency and medical departments. Furthermore, using a logistic model, we found no differences in contracting SARS-CoV-2 after the third dose based on the hospital department. Finally, analysis of antibody distribution following the spread of the omicron variant, subdividing the cohort of positive individuals into centiles, highlighted a cut-off of 550 BAU/mL and showed that subjects with antibodies below this are more susceptible to infection than those with a concentration above the established cut-off value.

## 1. Introduction

The extension and the duration of the immune response after infection or vaccination against severe acute respiratory syndrome coronavirus 2 (SARS-CoV-2) are relevant for implementing effective interventions against the pandemic, including the timing of vaccine boosters. In November 2021, worldwide, shortly before the spread of the omicron variant (B.1.1.529), the number of individuals affected was 3.39 billion people, corresponding to 44% of the population [1], suggesting that a large part of the world’s population has already experienced SARS-CoV-2 infection one or more times. The SARS-CoV-2 omicron variant has characteristics such as enhanced transmissibility and an ability to escape vaccine-derived immunity. COVID-19 vaccines have been developed with great efficiency. Although first-generation vaccines remained effective against severe disease and death, robust evidence on vaccine effectiveness (VE) and the durability of this protection against breakthrough reinfection by omicron variants, irrespective of symptoms, remains to be seen. Even in the case of infection against the SARS-CoV-2 virus, vaccination produced a protective response and the development of immune memory. The maximum protection against SARS-CoV-2 infection and severe COVID-19 disease in the population was observed in the presence of “hybrid” immunity (i.e., due to the combined effect of vaccination and previous infection) [1,2,3,4,5]. Over the last 4 years, numerous diagnostic tools have been developed for the quantitative and qualitative detection of the viral genome (RT-qPCR, RT-LAMP), as well as serological methods for identifying the antigenic proteins or antibodies of the virus. Several immunological assays have been developed for their detection in patient samples, such as anti-SARS-CoV-2 spike S glycoprotein RBD IgG, IgM or IgA antibodies (ELISA assay), the lateral flow assay (LFA), the chemiluminescent immunoassay (CLIA) and neutralization assays. The development of the humoral response with the production of antibodies represents an essential component of the reactivity of the immune system, and laboratory serological methods allow this response to be quantitatively detected and tracked over time. These methods specifically represent a useful tool for research purposes and in the epidemiological evaluation of viral circulation. Furthermore, they allow us to monitor the progress of the antibody response and to evaluate the effectiveness, duration and persistence of the protection it confers, even against reinfections [6,7,8].

Within the overall population, healthcare workers (HCWs) are the population segment most exposed to SARS-CoV-2 infection [9]. The quantification of antibody response persistence may also help to predict the effectiveness of vaccines against SARS-CoV-2. The antibodies triggered by infection and/or by vaccination show the entire pattern of the specific immunoglobulins produced. It is also known that vaccination has a main role in cross-immunity against coronaviruses, highlighting the importance of booster doses against infections caused by new viral variants and subvariants [10].

Antibodies binding to the SARS-CoV-2 spike protein, especially those neutralizing antibodies, correlate with protection [11]. Higher levels of these antibodies with neutralizing activity have been associated with increased protection in several studies based on both mRNA and viral vector vaccines, but there is no threshold at which a subject is considered to be protected. Moreover, although individuals can have neutralizing antibodies, they can still become reinfected. To assess the strength of protection, many other factors should be considered, such as different neutralizing antibody thresholds, or even different immune responses against severe, symptomatic or asymptomatic disease [11,12].

## 2. Materials and Methods

### 2.1. Characteristics of the Study Population

COVID-19-positive and -negative healthy workers (HCs) from the Hospital Pio XI of Desio, Italy, were included in the study (ABCV-Brianza, antibodies-Covid-Vaccino-Brianza), comprising 181 females (78%) and 51 males (22%) with a mean age of 51.9 years (range 23–69 years, median 54 years). None of the subjects studied were taking corticosteroids, and they were in good health. COVID-19 positivity was diagnosed by RT-PCR post nasopharyngeal swab. When the assay showed a weakly positive result, the RT-PCR analysis was repeated with another analytical instrument to confirm the result [1]. The result of RT-PCR was always associated with clinical symptoms. All the phenotypic variables, the hospital area of provenience and the starting antibody levels (T0) of the subjects are reported in Table 1.

The HCW subjects enrolled in the study received the first two doses of the Comirnaty vaccine (BNT162b2 vaccine, Pfizer–BioNTech, manufactured in Mainz, Germany) in compliance with the protocol followed (priming dose and booster dose, 30 μg mRNA with 0.3 mL/dose). The 2nd dose was administered exactly 21 days after the priming dose during the period from January 2021 to February 2021. The third dose (booster dose) was administer in November 2022.

Sequential serum samples were collected to measure anti-SARS-CoV-2 antibodies at the following time points: T0, before the priming dose (day 0); T1, before the 2nd dose (day 21); T2, exactly 14 days after the 2nd dose (day 35); T3, 4 months after the 2nd dose (day 120); T4, 9 months after the 2nd dose (day 240); T5, 12 months after the 1st dose and 2 months after the 3rd booster dose; T6, 17 months after the 1st dose and 7 months after the 3rd booster dose (Figure 1).

The time point T5 corresponded with the spread of the omicron variant and its subvariants (VOCs). Vaccine effectiveness, in terms of antibody response and infection-acquired immunity, were assessed. 

The quantitative measurement of total antibodies (including IgG) against the SARS-CoV-2 spike (S) protein RBD in human sera was performed with the Roche Elecsys anti- SARS-CoV-2 S immunoassay on the Roche Cobas c8000 platform (Roche Diagnostics GmbH, Mannheim, Germany). The interval assay was 0.4–250 BAU/mL; a positive test result was >0.80 BAU/mL and a negative result was <0.8 BAU/mL. Samples with a concentration > 250 BAU/mL were diluted (1:100) in accordance with the producer indication for values up to >25,000 but not beyond. 

The study was performed in accordance with the Declaration of Helsinki, and all participants gave informed consent. The study was approved by the local ethical committee.

### 2.2. Statistical Analysis

We initially evaluated the changes in the antibody response by quantification at different times. Since the antibody concentration distribution was not normal, we used the Friedman rank sum test to evaluate the significance of the differences among the groups. Significance was set at *p* < 0.05. In case of significant differences, we employed a nested Wilcoxon rank sum test for paired data for cross-comparison and to identify which specific time points or intervals differed from each other. The *p*-values were thus corrected with the False Discovery Rate (FDR) via multiple testing correction. Significance was considered for associations with *p*-values < 0.05. Then, we quantitatively monitored the variation in antibody response at the different time points while accounting for sex, age and hospital area. For this purpose, we applied the General Estimating Equation (GEE) model to adjust for potential unrecognized interdependencies among sequential measurements for the same individual. To reduce the impact of outliers and homogenize the distribution of the antibody response, in this model, we set the logarithmic antibody response as the outcome variable and included measurement times, sex, age and hospital department as covariates. Initially, we employed the model using the baseline inclusion status (T0) as a reference time to monitor initial changes. Later, we referenced all other time points to evaluate the significance and quantify associations across all the time points. Even when changing the reference time in our analyses, through the intrinsic characteristics of the model we adopted, we considered the initial antibody levels (T0). This ensured that any observed effects accounted for the initial logarithmic antibody levels. This approach helped us to avoid potential bias related to the antibody status at the time of inclusion in the study. Furthermore, having introduced the hospital department as a fixed-effect covariate to assess the variation in the average antibody response based on the hospital department, we changed the reference department of origin to evaluate the significance and quantify the average logarithm of the antibody response from department to department.

Finally, we focused on the phase post third booster dose, analyzing whether there was a relationship between contracting SARS-CoV-2 based on the results of the antigenic swab and the hospital area of provenience. We used a logistic regression model, for which we set the swab result as the response variable (yes or no) and the hospital area, sex and age as covariates. Subsequently, we calculated the odds ratios (ORs) and considered associations with a *p*-value < 0.05 as significant.

## 3. Results

Initially, the investigation using the Friedman rank sum test was significant with a *p*-value < 0.01, rejecting the null hypothesis of equal time points, thus indicating differences between at least one of the analyzed time points. Subsequent post hoc testing utilizing pairwise comparisons with the Wilcoxon rank sum test, including FDR correction for multiple testing *p*-values, revealed that all time points differed significantly from each other, with each comparison yielding a *p*-value < 0.01. Further analysis was carried out to quantitatively evaluate the variation in antibody levels and its significance concerning all covariates, including sex, age and hospital area. The analysis also showed significant variations across all time points. The results are summarized in Table 2, displaying the beta coefficients for the logarithm of antibodies with significance denoted by an asterisk. 

Detailed information, such as 95% confidence intervals, the standard error and interactions of the antibody response with all other covariates can be found in the Appendix A. Additionally, the logarithmic trend of antibodies is visually represented in Figure 2, with Panel A showing all subjects combined and Panel B depicting the differences for each medical department. As shown in Figure 2B, the department that stood out the most was the emergency department, though the administrative and surgery departments showed a very similar trend. 

From Table 3, we note that the average variation in the logarithm of the antibodies significantly differed across all departments except between the administrative and surgery departments, where no significant differences were present.

The logistic model implemented to assess the significant relationship between the presence of COVID-19 after the third dose and the department of origin did not yield any significant associations, as shown in Table 4 (see also Appendix A). 

In Figure 3, the values of the logarithm of antibodies for each hospital area in relation to the COVID-19 swab outcome are depicted. We observed that at both T5 and T6, the emergency department showed a lower number of antibodies compared to the other departments, as also observable in Figure 2. However, the trend of the antibody logarithm in relation to having contracted COVID-19 or not was similar across all four hospital areas. Specifically, at T5, there were slight differences between those who had contracted the infection and those who had not across all hospital departments. At T6, there was a greater increase in antibodies among those who had contracted SARS-CoV-2 compared to those who had not within the departments (see Figure 4).

Then, we divided the 232 subjects according to T0 positivity (before the first dose, with a value > 0.80 BAU/mL) into two groups: one with 198 subjects that were negative at T0 (i.e., negative) and one with 35 subjects that were positive at T0 (i.e., positive). Of the 198 subjects negative at T0 (79.8%), a subgroup of 60 subjects (30%) were positive for COVID-19 between January 2021 and July 2022, while 138 individuals (70%) remained negative (see Table 5).

Of the 60 positive subjects, 57 (95%) had a positive swab in the period of the omicron variant. We do not have the date of the swab test for the other three positive subjects. Of the 35 positive subjects at T0, 27 (54%) were positive between March 2020 and November 2020 before the first dose; 7 subjects (14%) tested positive again between November 2021 and February 2022, during the omicron variant period (Table 3). However, regarding the division of the subjects studied into groups according to COVID-19 positivity, we did not carry out a sequencing analysis of the viral genome.

We divided the subjects according to the date of the positive swab test, and we correlated this to the VOCs circulating in that period, which was determined to be the variant most likely to be contracted at that time. Then, we divided the cohort of positive individuals into centiles, based on a selected threshold/cut-off, corresponding to the value at which the antibody response of subgroups diverged. This cut-off was useful for discriminating HCWs who became infected with the omicron variant from those who did not. Those with the lower levels of antibodies had the omicron variant infection; we found that subjects with antibody levels between 400 and 550 BAU/mL were more susceptible to infection than those with concentrations > 550 BAU/mL. Therefore, we verified the hypothesis regarding whether the differences in levels above and below the threshold of 550 BAU/mL lead to a difference in terms of infection frequency: subjects with antibody levels < 550 BAU/mL (38 omicron-positive and 60 omicron-negative, 38/98) were reinfected by the variant in 40% of cases. Subjects with antibody levels > 550 BAU/mL (15 omicron-positive; 68 omicron-negative (15/83) contracted the variant in 18% of cases (Figure 5).

## 4. Discussion

Epidemiological evidence has highlighted that SARS-CoV-2 persists via its constant mutation and the spread of variants capable of triggering the process that can lead to circumventing the protective effect due to the neutralizing antibodies produced following a natural infection and/or vaccination or administered during passive immunotherapy. This is called vaccine-elicited antibody neutralization, infection-elicited antibody neutralization or both [13]. The degree of protection was particularly higher for those with “hybrid immunity” compared to those with a previous infection alone, reinforcing the importance of vaccination despite a previous infection to protect against severe disease due to the omicron variant. These observations should be interpreted cautiously; furthermore, we consider some groups to be at greater risk, such as the elderly and immunocompromised subjects including organ transplant recipients, for whom even the third booster dose may not be sufficient for protection [14,15,16]. With the analysis of the antibody response of 232 subjects to estimate how vaccination history modulated the risk of infection before and after the spread of the omicron variant, we observed differences existing in term of protection toward reinfection with the omicron variant. In accordance with the literature from the last 3 years, we consider the analysis of the antibody response trend over the time as the best indicator of the real protective effectiveness of the vaccines developed against COVID-19. While vaccines are frequently assessed by their ability to stimulate neutralizing antibodies, they can also induce antibodies with non-neutralizing functions critical to disease mitigation. It was shown that vaccine platform-induced humoral responses have distinct peak immunogenicity and waning profiles. These responses may be rescued and expanded with subsequent homologous and heterologous vaccination [13,17]. As shown in Figure 2, a significant reduction in the antibody response was observed as early as 4 months after the administration of the first two doses of the vaccine (T3), which was maintained after 8/9 months (T4). We did not have an available objective and stringent criterion to determine the risk of exposure, as we have seen cases even in which personal protective equipment has been rigorously used, and in sectors considered to be at a lower risk, there have been severe cases. We divided the groups of workers by biological risk: those at greatest risk were considered to be those working in the emergency area and internal medicine. Our results suggest that immunological memory in terms of antibody response was acquired in most individuals analyzed, both in those with a previous infection and in those only vaccinated, and it remained significantly high in the majority of subjects up to 9 months after vaccination, with a declining trend (T0–T4). This decline is well compensated for by an increase after the third dose (T5–T6). As is known, the entry of the virus into human cells is made possible by the interaction and binding of the receptor-binding domain (RBD) of the spike protein with ACE2, and at the same time, it constitutes the main target of the vaccines developed. Antibodies specific to the RBD portion of the spike protein are neutralizing antibodies that develop in response to natural infection or vaccination with protective immunity against the virus. The main utility of serological tests is to measure and produce all classes of immunoglobulin isotypes (IgG, IgA, IgM) to monitor the duration of the immune response in both infected (symptomatic or not) and vaccinated subjects. The rapid and robust recall of humoral immune responses observed after the third booster dose indicated that the primary two-dose vaccination regimen establishes a sustained immune memory. In previous work, no significant correlation between the antibody titer and gender or age was observed. Instead, a correlation was found between the type of vaccine and antibody response time [18]. In our study, the subjects were vaccinated with the same type of vaccine, so we cannot attribute this variation to the vaccine used. Hartley et al. showed that serum antibody levels decrease following antigen clearance as part of the response of the immune system, but memory B cells persist, which are capable of recognizing the virus and reactivating antibody production. Therefore, the decrease in serum antibodies from T2 to T4 that we noted in our study is in line with the antibody kinetics studied in other scientific works [19].

We also evaluated the differences between antibody levels in healthcare workers who were infected with the omicron strain (in blue) in comparison with who were not (in red) (Figure 5), investigating whether the differences in levels above and below a chosen threshold of 550 BAU/mL could indicate a difference in terms of the infection frequency of antibody levels. Starting from January, we found that the fraction of reinfected individuals was significantly higher for those with antibody levels < 550 BAU/L (Figure 5). In a systematic review published by Bobrovitz et al., it was highlighted that the combination of antibodies generated by anti-COVID-19 vaccination and recovery from the infection, also known as hybrid immunity, would offer greater protection against severe forms of COVID-19 and resulting hospital admission. In particular, the review showed that protection against severe disease and hospitalization remains high 12 months after developing immunity, compared to being unvaccinated and uninfected. This author showed that the probability of contracting severe COVID-19 or needing hospitalization one year after developing hybrid immunity is at least 95% lower, while in people infected a year earlier, but who are not vaccinated, the risk is 75% lower, highlighting that protection against reinfection was lower than that against severe disease, with people with hybrid immunity having a 42% lower chance of being reinfected with the coronavirus one year later, while those had only been infected have a 25% lower risk [10]. The phenomenon described in Figure 5 is, at least in part, the result of the so-called antigenic sin, i.e., the propensity of the human immune system to use immunological memory instead of recreating new antibodies following a second exposure to the pathogen, even if it has different characteristics from the original one. During primary and secondary infections or following vaccination, a virus can undergo antigenic changes, a process characterized by natural mutational events of epitopes that could evade the protection systems designed by the immune system, despite the activation of memory B cells. This can happen for several reasons: (a) antibodies produced by memory B cells fail to bind to altered epitopes and (b) these antibodies inhibit the activation of new virgin B lymphocytes, which would allow the generation of more efficient antibodies against the evolved pathogen.

The greater transmissibility and immunoevasion capacity of the omicron variant and its subvariants have generated several successive waves of new cases since the first omicron case identified in November 2021, with the omicron variant BA.1 accounting for 90% of sequenced infections by January 2022. Since then, the omicron subvariants BA.1, BA.2 and BA.5 have emerged [20,21,22]. Several age-independent host-related factors such as gender, or the preservation of immunocompetence with controlled inflammation during antigenic challenges, which is a hallmark of immunoresilience could explain the varying risks for severe coronavirus disease infection/reinfection. A recent Israelian study showed that BNT162b2, a homologous booster dose, was associated with a lower rate infection rate [23,24,25]. A critical aspect of the study is the Roche assay that was used, as it detects the concentration of total anti-protein S RBD antibodies without discrimination between the different isotypic classes of immunoglobulins secreted. Finally, a limit of our study is the lack of evaluation of other compartments of the immune system, in particular cell-mediated immunity, which would certainly provide more exhaustive and detailed results on the overall immune response of the subjects studied and which, in a coordinated manner, is responsible of immunological memory. As well as the role highlighted in the literature of abnormally high levels of the IgG4 subclass detected after repeated vaccination with an mRNA vaccine, which could not be a protective mechanism but rather a tolerogenic one to the spike protein, this could promote an unchallenged reinfection and viral replication.

## 5. Conclusions

One of the characteristics of this study was the follow up of the subjects for 17 months to monitor the antibody response and the risk of reinfection with the omicron variant, confirming what emerged from the literature data on the effectiveness of the vaccine; however, its duration is not very long-lasting, as demonstrated by the decrease in antibody levels from T2 to T4 and the subsequent increase after the booster (T5, T6). The response of subjects who were infected before vaccination was greater than that of subjects who were only vaccinated. This work highlights that the variations in the humoral response following natural infection and vaccination of a cohort of healthcare workers followed for over a year and a half are substantially stable, although with slight decreases from T2 to T4, followed by an increase after the administration of the three booster doses (T5–T6), confirming the good effectiveness of the mRNA vaccine used in our healthcare setting.

## Figures and Tables

**Figure 1 vaccines-12-00506-f001:**
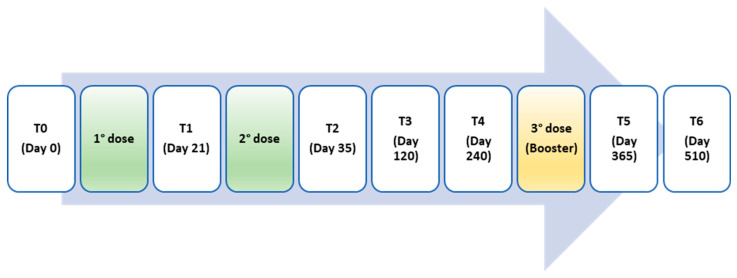
Time points of vaccine administration and serum collection.

**Figure 2 vaccines-12-00506-f002:**
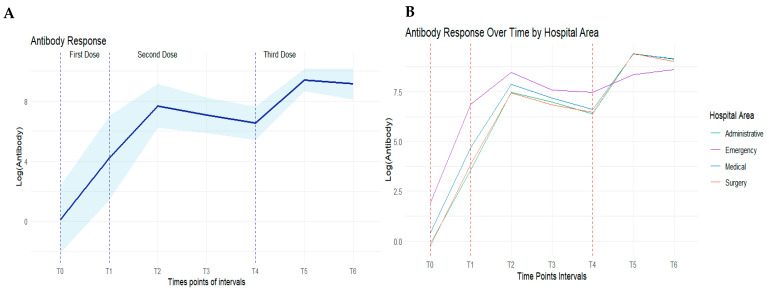
Panel (**A**) Boxplot of the kinetics of the humoral antibody response in all hospital staff. Panel (**B**) Boxplot of the kinetics of the antibody response by hospital area. In both charts, T0 indicates the work confinement phase, T1 the measurement at the first dose, T2–T4 the interval during the second dose and T5–T6 the interval of the third dose.

**Figure 3 vaccines-12-00506-f003:**
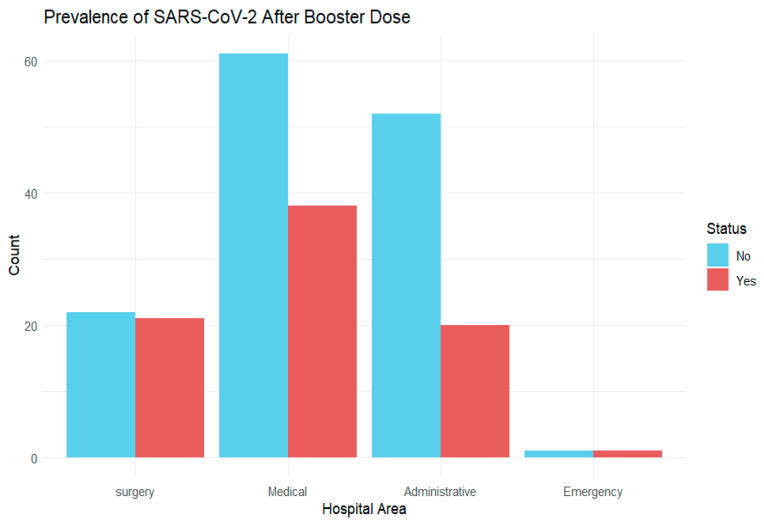
Barplot displaying the prevalence of SARS-CoV-2 infection after booster dose in HCWs by hospital area.

**Figure 4 vaccines-12-00506-f004:**
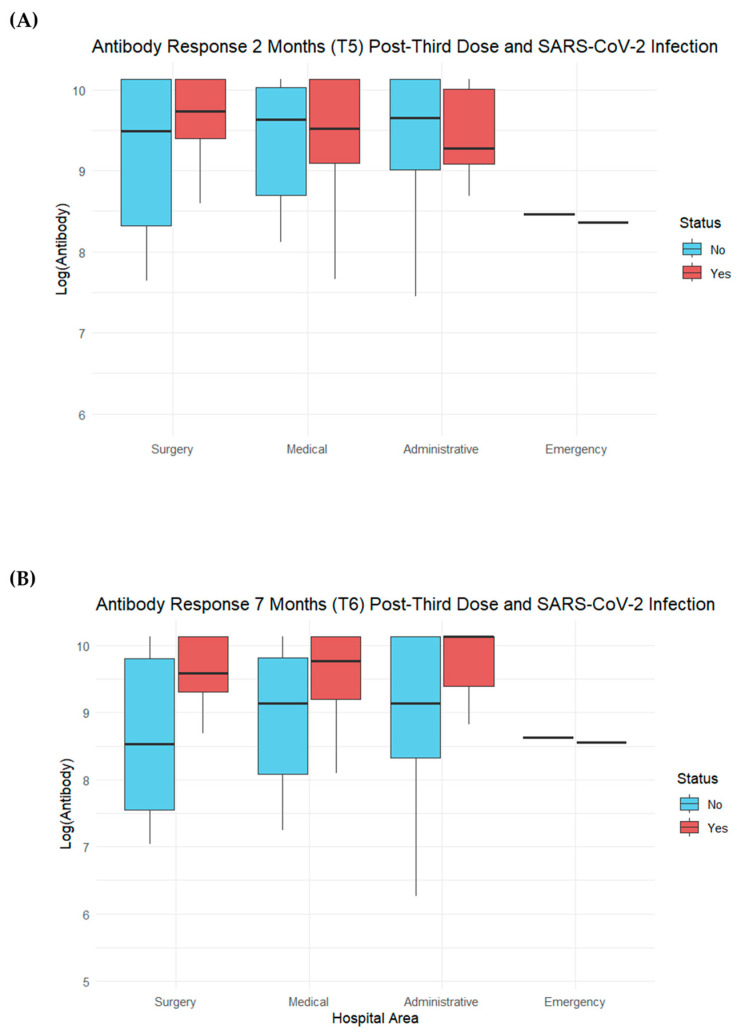
Boxplot depicting log antibody values by hospital department in relation to the COVID-19 swab outcome (yes or no). In Panel (**A**), we see this relationship 2 months after the booster dose, while in Panel (**B**), this relationship appears 7 months after the third dose.

**Figure 5 vaccines-12-00506-f005:**
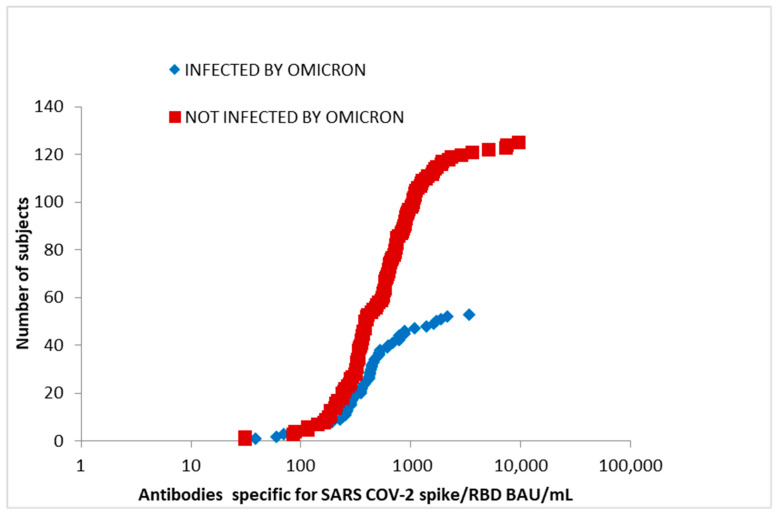
Cumulative distribution of antibody levels in HCWs disaggregated into those who were infected with omicron (in blue) and those who were not infected (in red).

**Table 1 vaccines-12-00506-t001:** Phenotypic variables of subjects included in the study, presented as overall values and then values for various departments. For continuous variables such as antibody response and age, the mean, respective standard deviation and median values with respective range [min, max] are provided. For categorical variables such as gender and whether or not the cut-off threshold of the immunoassay was exceeded, the prevalence percentages are shown.

	Medical	Administrative	Emergency	Surgery	Overall
Hospital Area Prevalence	45.49%	32.37%	2.04%	20.08%	
Antibody (BAU/mL)					
Mean (SD)	156 (1130)	52.6 (246)	48.8 (84.9)	26.3 (124)	94.2 (780)
Median	0.400	0.400	13.0	0.400	0.400
[Min–Max]	[0.400–11,900]	[0.400–1990]	[0.400–199]	[0.400–837]	[0.400–11,900]
Immunoassay results > 0.8					
No	73.9%	86.1%	40.0%	87.8%	79.9%
Yes	26.1%	13.9%	60.0%	12.2%	20.1%
Sex					
F	85.6%	70.9%	80.0%	69.4%	77.5%
M	14.4%	29.1%	20.0%	30.6%	22.5%
Age					
Mean (SD)	52.0 (9.47)	52.1 (8.34)	56.9 (6.75)	51.5 (9.79)	52.0 (9.12)
Median	53.6	54.3	58.0	54.4	54.2
[Min–Max]	[25.3–69.8]	[32.9–71.7]	[47.1–65.1]	[26.1–64.7]	[25.3–71.7]

**Table 2 vaccines-12-00506-t002:** Summary of beta coefficients, obtained from the GEE models, representing an increase or decrease in the antibody response on the logarithmic scale across all time points. The significance of the association (*p* < 0.05) is indicated by an asterisk (*) placed next to the value. C.I. 95% and S.E. are reported for each model in the Appendix A.

	T1	T2	T3	T4	T5	T6
T0	4.08 *	7.55 *	6.93 *	6.40 *	9.30 *	9.03 *
T1	-	3.46 *	2.84 *	2.31 *	5.22 *	4.94 *
T2	-	-	−0.62 *	−1.15 *	1.75 *	1.47 *
T3	-	-	-	−0.53 *	2.37 *	2.09 *
T4	-	-	-	-	2.90 *	2.63 *
T5	-	-	-	-	-	−0.27 *

**Table 3 vaccines-12-00506-t003:** Summary of the average variation in the logarithm of antibodies based on the hospital department as a fixed effect in the GEE model. The table reports the beta coefficient, which represents the increase or decrease in the logarithm of antibodies from one department to another, the S.E. (standard error) and the 95% C.I. (confidence interval). Significant associations (*p* < 0.05) are highlighted with an asterisk (*).

	Medical	Administrative	Emergency	Surgery
Medical	-	[−0.34; 0.09; (−0.53, −0.16)] *	[1.08; 0.39; (0.31, 1.85)] *	[−0.35; 0.12; (−0.57, −0.12)] *
Administrative	-	-	[1.42; 0.39; (0.65, 2.2)] *	[0; 0.11; (−0.22, 0.22)]
Emergency	-	-	-	[−1.43; 0.4; (−2.21, −0.64)] *

**Table 4 vaccines-12-00506-t004:** Logistic model results showing the relationship between SARS-CoV-2 infection status after third dose with various hospital areas and the other covariates. Reference level used for categorical variables (hospital area employment and gender) are denoted as “Reference”.

SARS-CoV-2 Status	Beta Coefficient	Standard Error	*p*-Value	Odds Ratio	C.I. 95%
Medical	Reference	-	-	-	-
Emergency	0.51	1.43	0.71	1.67	(0.10, 28.01)
Surgery	0.45	0.37	0.22	1.57	(0.75, 3.30)
Administrative	−0.49	0.34	0.14	0.60	(0.30, 1.19)
Age	−0.02	0.01	0.06	0.97	(0.94, 1.00)
Sex (Female)	Reference	-	-	-	-
Sex (Male)	−0.13	0.37	0.72	0.87	(0.42, 1.81)

**Table 5 vaccines-12-00506-t005:** Population quartile: median, 1st–3rd quartile and [min–max] intervals.

	Negative at T0/ Always Negative *n* = 138 (59%)	Negative at T0/ Positive after 2nd Dose and Omicron Reinfection *n* = 60 (26%)	Positive at T0/ Positive after Omicron Reinfection *n* = 7 (3%)	Positive at T0/ Only Positive before the 1st Dose *n* = 27 (12%)
**T0**	<0.4 BAU/mL	<0.4 BAU/mL	16	69.5
			1–330.5	18–386
			[1–983]	[3–1193]
**T2**	1638	1401	5897	13576.5
	849–2828.25	785–2955.5	3411–25,000	9911–25,000
	[0.4–12,571]	[0.4–11,958]	[1091–25,000]	[2945–25,000]
**T4**	569	427	1159	1643
	320–909.5	270.5–783	483.25–8622.5	945–3614
	[31–9557]	[39–25,000]	[397–8812]	[535–9382]

## Data Availability

The authors confirm that the data used for the findings in this study will be made available through the corresponding authors to qualified and interested investigators upon reasonable request.

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
