# Peer review of "Antibody Response against SARS-CoV-2 after mRNA Vaccine in a Cohort of Hospital Healthy Workers Followed for 17 Months"

_vaccines, 2024, doi:10.3390/vaccines12050506_

Round 1

Reviewer 1 Report

Comments and Suggestions for Authors

Dear Editor, Dear Authors,

Thank you for the opportunity to review the manuscript entitled “Antibody response against SARS COV-2 after mRNA vaccine in a cohort of hospital healthy workers followed for 17 months”

The manuscript addresses some interesting topics associated with the immunological response of health care workers after the exposure to SARS-CoV-2 or its antigens. The study is quite interesting, however, in my opinion, the scientific soundness of the manuscript should benefit considering the following issues:

*Introduction

-page2-line79: the comment states rather about effectiveness instead of efficacy [consider please “Efficacy can be defined as the performance of an intervention under ideal and controlled circumstances, whereas effectiveness refers to its performance under 'real-world' conditions”].

*Materials and Methods

-provide some data supporting the representativeness of the study sample,

-to get reliable results the authors should assure the accuracy of the end-point identification. It should be clearly described what were the criteria to identify the presence of SARS-CoV-2 among participants during the study period. If this was also RT-PCR post nasopharyngeal swab, how the authors dealt with false-positive results [see “False-Positive Results of SARS-CoV-2 RT-PCR in Oropharyngeal Swabs From Vaccinators” doi: 10.3389/fmed.2022.847407]

*Statistical analysis

-I suggest considering repeated-measures analysis of variance and the growth mixed models to evaluate broader the nature of differences of changes in antibody levels over time;

-there is an inconsistency in the statement about Bonferroni correction, as the correction yields to the decrease in the error rate which is considered as a cut-off for statistically significant results. Therefore keeping “We considered comparisons significant with a p-value <0.05” is somehow mistaken.

*Results

-there is no background characteristic of the study sample (more than just gender and age). I believe it is reasonable to present participants by their COVID-19 positive/negative status at the inclusion to the study

-under introduction authors highlight the effect of hybrid immunity, however there is no analysis which addresses that issue even if the authors have the data. Were there any chances in the antibody response between COVID-19 positive/negative groups?

-there is no information about ‘being in contact with COVID-19’ during the observation period. This is however, among factors determining the level of antibody response;

-page.3-line.141: the statement ‘indicate a significant association between antibody kinetics and time intervals’ is to some degree mistaken, as the analysis shows just the difference in log antibody at different time points but not the differences in the kinetics across time intervals.

-for Tab.1 analysis among important covariates which should be used are working hospital area and SARS-CoV-2 status at the beginning of the study;

-page.3-line.148, the sentence refers to effectiveness rather than efficacy;

-Tab.1 p-value: probably there is a typing error and ‘e’ should not be placed as an index of the power;

-Tab.1: next to beta coefficients present also standard errors

-Fig.2; my comment about prevalence determination see before. Second is that it would be more informative to present the cumulative prevalence over time – to have a possibility to relate it to the antibody status in the group(s);

-Tab.2: present ORs with 95%CI from logistic models rather, than just bet coeff.

-Tab.3 is poorly described. Provide in the title, what are the groups and the measurements presented, and explain signs as ‘*’. Decimal points should be corrected;

*Discussion:

-regarding antibody response, leave a comment about how occupational exposure might have an impact on antibody kinetics;

-study limitations should be considered and discussed more broadly;

*Conclusion (manuscript main body)

-overall, the statement is a generalization of what is currently known. Make a conclusion which refers to your results regarding antibody kinetics and risk of infection;

-the study investigated an effectiveness rather, but not efficacy.

Best, Reviewer

Comments on the Quality of English Language

English require to be reviewed and corrected by the native speaker, especially grammar under Abstract and Materials and Methods.

Author Response

REVIEWER 1

Dear Editor, Dear Authors,

Thank you for the opportunity to review the manuscript entitled “Antibody response against SARS COV-2 after mRNA vaccine in a cohort of hospital healthy workers followed for 17 months”

The manuscript addresses some interesting topics associated with the immunological response of health care workers after the exposure to SARS-CoV-2 or its antigens. The study is quite interesting, however, in my opinion, the scientific soundness of the manuscript should benefit considering the following issues:

*Introduction

-page2-line79: the comment states rather about effectiveness instead of efficacy [consider please “Efficacy can be defined as the performance of an intervention under ideal and controlled circumstances, whereas effectiveness refers to its performance under 'real-world' conditions”].

Done. Please see line 70 and followings.

*Materials and Methods

-provide some data supporting the representativeness of the study sample

Done

-to get reliable results the authors should assure the accuracy of the end-point identification. It should be clearly described what were the criteria to identify the presence of SARS-CoV-2 among participants during the study period. If this was also RT-PCR post nasopharyngeal swab, how the authors dealt with false-positive results [see “False-Positive Results of SARS-CoV-2 RT-PCR in Oropharyngeal Swabs From Vaccinators” doi: 10.3389/fmed.2022.847407]

We used RT-PCR post nasopharyngeal swab. When the result was weakly positive we repeated the RT-PCR analysis with another analytical instrument to confirm the result. Furthermore the result of RT-PCR was associated with clinical symptoms. Phenotype table was introduced where the initial antibody values ​​were reported.

We report some points mentioned in the article “False-Positive Results of SARS-CoV-2 RT-PCR in Oropharyngeal Swabs From Vaccinators” doi: 10.3389/fmed.2022.847407”:

“…Furthermore, employees and vaccine injectors are required not to perform SARS-CoV-2 nucleic acid testing within 48 hr after vaccination, to avoid false-positive caused by vaccine contamination..”;

“…In conclusion, the detection of viral RNA in nasopharyngeal/oropharyngeal swabs is the gold standard for the diagnosis of COVID-19..”

*Statistical analysis

-I suggest considering repeated-measures analysis of variance and the growth mixed models to evaluate broader the nature of differences of changes in antibody levels over time;

Friedman test is the non parametric equivalent of the required test. The GEE which is the test carried out satisfies this comment because it has now been repeated by changing the reference times in order to reconstruct the progress of the vaccination response.

-there is an inconsistency in the statement about Bonferroni correction, as the correction yields to the decrease in the error rate which is considered as a cut-off for statistically significant results. Therefore keeping “We considered comparisons significant with a p-value <0.05” is somehow mistaken.

The previous mistake was not specifying that the p-value was adjusted and not nominal. So we taked an adjusted <0.05 not a nominal <0.05. The correction used is false discovery rate (FDR) now which is a little less restrictive and changes the parameter mentioned less.

*Results

-there is no background characteristic of the study sample (more than just gender and age). I believe it is reasonable to present participants by their COVID-19 positive/negative status at the inclusion to the study see phenotypic Table 1.

-under introduction authors highlight the effect of hybrid immunity, however there is no analysis which addresses that issue even if the authors have the data. Were there any chances in the antibody response between COVID-19 positive/negative groups?

We have no analysed the hybrid immunity of our cohort and but the results we observed are overlapping with those cited in the work from Bobrovitz et al.  Protective effectiveness of previous SARS-CoV-2 infection and hybrid immunity against the omicron variant and severe disease: a systematic review and meta-regression Niklas Bobrovitz, Harriet Ware, Xiaomeng Ma, Zihan Li, Reza Hosseini et al. www.thelancet.com/infection  Vol 23 May 2023, 556-567.

-there is no information about ‘being in contact with COVID-19’ during the observation period. This is however, among factors determining the level of antibody response;

We evaluated only the result of molecular swab.

-page.3-line.141: the statement ‘indicate a significant association between antibody kinetics and time intervals’ is to some degree mistaken, as the analysis shows just the difference in log antibody at different time points but not the differences in the kinetics across time intervals.

Done

-for Tab.1 analysis among important covariates which should be used are working hospital area and SARS-CoV-2 status at the beginning of the study.

-page.3-line.148, the sentence refers to effectiveness rather than efficacy;

Done.

-Tab.1 p-value: probably there is a typing error and ‘e’ should not be placed as an index of the power;

Done

-Tab.1: next to beta coefficients present also standard errors.

Done

-Fig.2; my comment about prevalence determination see before. Second is that it would be more informative to present the cumulative prevalence over time – to have a possibility to relate it to the antibody status in the group(s);  

Done

-Tab.2: present ORs with 95%CI from logistic models rather, than just bet coeff.

Done.

-Tab.3 is poorly described. Provide in the title, what are the groups and the measurements presented, and explain signs as ‘*’. Decimal points should be corrected;

Done

*Discussion:

-regarding antibody response, leave a comment about how occupational exposure might have an impact on antibody kinetics;

Done

-study limitations should be considered and discussed more broadly;

A limit of this study is the lack of evaluation of cell mediated immunity, which would certainly provide more exhaustive and detailed results on the overall immune response of the subjects studied. As well as the role  highlighted in the literature regarding the role of  abnormally high levels of  IgG4 subclass detected after repeated vaccination with mRNA vaccine, which could not be a protective mechanism but rather a tolerogenic one to the spike protein, that could promote an unchallenged reinfection and viral replication.

IgG4 Antibodies Induced by Repeated Vaccination May Generate Immune Tolerance to the SARS-CoV-2 Spike Protein. Review Vladimir N. Uversky, Elrashdy M. Redwan, William Makis and Alberto Rubio-Casillas.

Done

*Conclusion (manuscript main body)

-overall, the statement is a generalization of what is currently known. Make a conclusion which refers to your results regarding antibody kinetics and risk of infection;

One of the aspects of the study on the cohort that differentiates it from other works is that of having carried out a follow up of the subjects for up to 17 months and confirmed what emerged from the literature data on the effectiveness of the vaccine although its duration is not very long lasting, as demonstrated by the decrease from T2 to T4 and the subsequent increase in antibody levels after the booster (T5, T6). The response of subjects who were infected before vaccination is greater than that of subjects who were only vaccinated. Furthermore, by subdividing the groups on the basis of an arbitrary threshold value of antibodies (550 BAU/mL), obtained with the method in use, it was seen that subjects with values lower than the threshold had a greater risk of reinfection from the omicron variant than those with a higher value (Figure 4).

Done

-the study investigated an effectiveness rather, but not efficacy.

Done

 Comments on the Quality of English Language

English require to be reviewed and corrected by the native speaker, especially grammar under Abstract and Materials and Methods.

Linguistic revision was done.

Reviewer 2 Report

Comments and Suggestions for Authors

1.  The authors perform a longitudinal and prospective analysis of the anti-SARS-CoV-2 20 antibodies in a cohort of 232 healthy workers at multiple time points over a 17-months period to determine how circulating antibody levels change over time following natural infection and vaccination for SARS-CoV-2 before and after the spread of omicron variant, analyzing antibody response. However, it would have been interesting to have studied not only the antibody response but also the T cell-mediated immune response.

2. The last sentence of the Introduction section (lines 73 to 79). "With the analysis of antibody response of 232 subjects to estimate how vaccination histories modulated risk of infection before and after diffusion of omicron variant, we observed.." provides information that should be commented on in the Results and Discussion sections but not in the Introduction

3. With respect to the characteristics of the population studied, it is worth specifying that the healthy volunteers were examined and all had a normal physical examination, normal routine laboratory results, and none were taking corticosteroids or medications with an immunomodulatory effect.

4.  Their results did not reveal a significant association with the working area but only an association related to the age of the workers. However, the controls had an average age of 51.9 years and none were older than 69 years. Therefore, it should be 1.  Therefore, it should be taken into account that the results presented only apply to this age segment.

  Comments on the Quality of English Language

 Minor editing of English language required

Author Response

REVIEWER 2

Comments and Suggestions for Authors

1. The authors perform a longitudinal and prospective analysis of the anti-SARS-CoV-2 20 antibodies in a cohort of 232 healthy workers at multiple time points over a 17-months period to determine how circulating antibody levels change over time following natural infection and vaccination for SARS-CoV-2 before and after the spread of omicron variant, analyzing antibody response. However, it would have been interesting to have studied not only the antibody response but also the T cell-mediated immune response.

We have not studied T cell-mediated immune response.

 2. The last sentence of the Introduction section (lines 73 to 79). "With the analysis of antibody response of 232 subjects to estimate how vaccination histories modulated risk of infection before and after diffusion of omicron variant, we observed.." provides information that should be commented on in the Results and Discussion sections but not in the Introduction.

DONE

3. With respect to the characteristics of the population studied, it is worth specifying that the healthy volunteers were examined and all had a normal physical examination, normal routine laboratory results, and none were taking corticosteroids or medications with an immunomodulatory effect.

Done. Please see line 85.

4. Their results did not reveal a significant association with the working area but only an association related to the age of the workers. However, the controls had an average age of 51.9 years and none were older than 69 years. Therefore, it should be 1. Therefore, it should be taken into account that the results presented only apply to this age segment. 

Done

Comments on the Quality of English Language

 Minor editing of English language required

Reviewer 3 Report

Comments and Suggestions for Authors

Comments:

1. Line 16, replace “COVID-19” with SARS‐CoV‐2.

2. Line 30, cut-off of 550 BAU/mL, this is your cut-off or world consensus one? Please clarify it.

3. The manuscript estimated/measured antibody or antibodies? Which one IgGs/IgM/IgA? Mentioned which one (in the abstract).

4. From line 21 and 119 the authors estimated the total IgGs antibody only and not antibodies, so please correct that in line 21 and a similar position in the entire MS (such in figure 3 legend).

5. Lines 70-72, “…. recent studies ….” where are these studies (Refs), then the sentence is understandable, are you mean … by bivalent vaccines can induce these cross immunities? Please clarified the sentence/rephase

6. Line 75, replace (diffusion) by (distribute).

7. Your opinion in lines 77-79 seem reasonable and very good.

8. It is not clear why you jump back to first dose in T5/T6 and not calculate the days after similar to T0 to T4? To be more understandable and clearer try to draw line for primary, secondary the booster doses.

9. As the MS thoroughly mentioned that not all IgG induced after mRNA vaccination is protective (neutralizing), so who you conclude this in your MS. 

10. Then what is the explanation for the IgGs decline from T2 to T4? Many reports indicated it is due to the IgG4 production, which already can estimate as IgG (in total IgGs estimation)?

11. Some of your conclusion is risky such as females’ immune response was higher due you to the limited number of samples, try to rephrase this sentence.

Author Response

Comments:

1. Line 16, replace “COVID-19” with SARS‐CoV‐2.

Done. We have replaced “COVID-19” with SARS‐CoV‐2

2. Line 30, cut-off of 550 BAU/mL, this is your cut-off or world consensus one? Please clarify it.

We have decided arbitrarly the cut-off 550 BAU/mL. See Figure 3.

3. The manuscript estimated/measured antibody or antibodies? Which one IgGs/IgM/IgA? Mentioned which one (in the abstract).

We have measured total antibodies (See line 102, Materials and Methods)

4. From line 21 and 119 the authors estimated the total IgGs antibody only and not antibodies, so please correct that in line 21 and a similar position in the entire MS (such in figure 3 legend).

We have always estimated the total antibodies.

5. Lines 70-72, “…. recent studies ….” where are these studies (Refs), then the sentence is understandable, are you mean … by bivalent vaccines can induce these cross immunities? Please clarified the sentence/rephase

Done. Read line 71-73.

 6. Line 75, replace (diffusion) by (distribute).

Done.

 7. Your opinion in lines 77-79 seem reasonable and very good.

Thank you.

8. It is not clear why you jump back to first dose in T5/T6 and not calculate the days after similar to T0 to T4? To be more understandable and clearer try to draw line for primary, secondary the booster doses.

See figure 1

9. As the MS thoroughly mentioned that not all IgG induced after mRNA vaccination is protective (neutralizing), so who you conclude this in your MS.

Furthermore as described by Jennifer Abbasi, antibodies  binding  to the SARSCoV-2 spike protein particularly neutralizing antibodies, did correlate with protection. Higher levels of these antibodies were associated with increased protection in  several studies based on both mRNA and viral vector vaccines but there is no a simple relationship referred which antibody threshold should be considered protective, whereas individuals can have neutralizing antibodies but still get reinfected. Other more factors must be taken into account  to assess the strength of protection, such as  different neutralizing antibody thresholds, or even different immune responses against severe, symptomatic or asymptomatic disease.

Jennifer Abbasi The Flawed Science of Antibody Testing for SARS-CoV-2 Immunity. Medical News & Perspectives. JAMA November 9, 2021 Volume 326, Number 18, 1781-82.

10. Then what is the explanation for the IgGs decline from T2 to T4? Many reports indicated it is due to the IgG4 production, which already can estimate as IgG (in total IgGs estimation)?

We have not estimated the levels of IgG4 subclasses.

11. Some of your conclusion is risky such as females’ immune response was higher due you to the limited number of samples, try to rephrase this sentence.

Done.

Round 2

Reviewer 1 Report

Comments and Suggestions for Authors

Dear Authors,

The submitted manuscript has been clearly improved after the revision. Explanations considering statistical analysis and results added the content make the content more comprehensive to the potential readers.

Regarding some methodological issues, my minor comment is that statistical significance is like "yes/no: feature, meaning the result is /or is not statistically significant, and there is no grading in that concept [there is no such options like more or less, higher or lower, very high or very low statistical significance]. I suggest rewording the sentence "the least significant difference" (page 5 line 171-172).

Next, in Tab.3 cell 'surgery, medical' is marked as significant, however, 95% CI shows no significant result, so one explanation is that '-' (minus) has been mistakenly omitted ... or the result is not statistically significant.

Comments on the Quality of English Language

Minor editing of English is required, as there are some grammar error in the text. For example, see page 4 line 153 'biases' ... 'bias' is an uncountable noun.

Author Response

Dear Authors,

The submitted manuscript has been clearly improved after the revision. Explanations considering statistical analysis and results added the content make the content more comprehensive to the potential readers.

Regarding some methodological issues, my minor comment is that statistical significance is like "yes/no: feature, meaning the result is /or is not statistically significant, and there is no grading in that concept [there is no such options like more or less, higher or lower, very high or very low statistical significance]. I suggest rewording the sentence "the least significant difference" (page 5 line 171-172).

Next, in Tab.3 cell 'surgery, medical' is marked as significant, however, 95% CI shows no significant result, so one explanation is that '-' (minus) has been mistakenly omitted ... or the result is not statistically significant.

We agree with the Reviewer comments and we have modified the manuscript accordingly.

Comments on the Quality of English Language

Minor editing of English is required, as there are some grammar error in the text. For example, see page 4 line 153 'biases' ... 'bias' is an uncountable noun.

We worked to improve the English language quality.

Reviewer 3 Report

Comments and Suggestions for Authors

  1. Then what is the explanation for the IgGs decline from T2 to T4? Many reports indicated it is due to the IgG4 production, which already can estimate as IgG (in total IgGs estimation)?

We have not estimated the levels of IgG4 subclasses.

BUT YOU DO NOT RELEASE A RESNABLE DISSUSSION  for the IgGs decline from T2 to T4?

Author Response

Then what is the explanation for the IgGs decline from T2 to T4? Many reports indicated it is due to the IgG4 production, which already can estimate as IgG (in total IgGs estimation)?

We have not estimated the levels of IgG4 subclasses.

BUT YOU DO NOT RELEASE A RESNABLE DISSUSSION  for the IgGs decline from T2 to T4?

In other works, a IgG decline was described.

We modified the manuscript as following

“Serum antibody levels decline following antigen clearance as part of the contracting immune but that

memory B cells persist, capable of recognizing the virus and reactivating the production of antibodies.

  1. E. Hartley et al., Sci. Immunol. 10.1126/sciimmunol.abf8891 (2020)”